# Position: Behavioral Systems Require Behavioral Tests

**Manuel Cherep** [* 1]   **Nikhil Singh** [* 2]   **Pattie Maes** [1]

## Abstract

Artificial agentic systems increasingly operate as *behavioral* systems by interacting with dynamic environments, pursuing goals, and adapting over time. Yet, current evaluation methods largely focus on performance outcomes, not the underlying behavioral processes that produce them. This paper argues that AI agents must be evaluated like other behavioral systems: through systematic observation, perturbation, and interpretation of their actions. We draw on lessons from the behavioral sciences to motivate this position, and propose a research agenda focused on developing rigorous *behavioral tests*. These include methods for recovering decision strategies from action sequences, constructing environments that isolate behavioral differences, and probing emergent dynamics in multi-agent systems. Taken together, these directions offer a roadmap for developing a science of AI behavior.

## 1. Introduction

Imagine there are two people navigating similar negotiation scenarios and arriving at the same outcome, a favorable discount for their client. One gets there via persuasion: listening carefully to discover shared values and principles. The other relies on intimidation: issuing threats and applying pressure. To a casual observer, both have achieved the intended outcome. This is a classic case of *equifinality*, wherein the similarity in the endpoint can mask how different behavioral pathways led to it. In the behavioral sciences, this prompts deeper inquiry. Which strategies were employed? What environmental constraints drove them? Performance alone does not offer answers to these questions, but the answers can clearly be of great social consequence.

**This position paper argues that AI agent systems must be treated like other behavioral systems: their observable outputs must be interpreted through the lens of the processes that generate them.** This goal calls for the development of *a science of AI agent behavior*.

By *behavior* here, we mean the patterns of action that arise from the interaction between an agent and its environment (including other behavioral systems), often conditioned by objectives, constraints, inputs, and internal mechanisms. Studying behavior in this sense allows us to ask not just *what* an agent did, but *when* it does it, *why* it behaved that way under specific conditions, and *how* its behavior might change under others (cf. (Tinbergen, 1963)).

We argue that this effort demands new theory, methods, and engineering. While we can draw on the wisdom of established behavioral sciences such as psychology, cognitive science, ethology, and economics, it is also important to note that artificial agents differ from biological ones. They operate under different design constraints and different interfaces to the world. For example, we need principled methods for probing agent behavior across controlled environment variations, for inferring strategies from trajectories, and for constructing minimal tests that reveal failures and misalignments.

We position this agenda as distinct from, but complementary to, work in interpretability, formal verification, and benchmarking. Interpretability focuses on internal representations; verification deals with formal properties under pre-specified assumptions; benchmarking implements comparative performance on fixed tasks. A behavioral science of AI agents instead centers the relationship between systems and their environments, and provides a basis for explaining unexpected behaviors, comparing qualitatively different policies that achieve similar outcomes, and designing evaluation protocols sensitive to strategy. In this paper, we lay out the rationale for this research direction, and sketch a roadmap for practically realizing it.

### 1.1. The Rise of *Behavioral* AI Systems

To define behavioral AI systems, it is helpful to first contrast them with what they are not. Most widely deployed AI systems, such as image classifiers, spam filters, and credit risk models, can be characterized as *static*. They map inputs

---

[*]Equal contribution   [1]Media Lab, Massachusetts Institute of Technology, Cambridge, USA [2]Department of Computer Science, Dartmouth College, Hanover, USA. Correspondence to: Manuel Cherep <mcherep@mit.edu>, Nikhil Singh <nikhil.u.singh@dartmouth.edu>.

*Proceedings of the 43$^{rd}$ International Conference on Machine Learning*, Seoul, South Korea. PMLR 306, 2026. Copyright 2026 by the author(s).

to outputs in a one-shot fashion and are typically stateless. These systems serve as decision aids rather than autonomous actors. In this setup, the human interprets the output and decides how to act. Evaluation focuses on input-output correctness metrics, e.g. classification accuracy and error types for a classifier.

Behavioral AI systems, by contrast, are *dynamic*. They interact with non-stationary environments, making sequences of decisions based on observations over time. Their behavior is shaped by intermediate outcomes, which may influence future actions. This class includes RL agents, many robotic systems, and now modern LLM agents embedded in tool-augmented environments such as the web (Nakano et al., 2021; Zhou et al., 2023; Koh et al., 2024), operating systems (Mialon et al., 2023; Kim et al., 2024), and financial platforms (Yu et al., 2024). We focus in this paper on this last category of LLM-based agents, because we believe they are most drastically underserved by today's AI evaluation infrastructure. Unlike static models, these agents generate open-ended action sequences conditioned on environment state. Their behavior cannot be exhaustively predicted from their design, and must instead be characterized through observation, perturbation, and interpretation.

Agent-based systems are not new. Early versions appeared under names like "software agents," "interface agents" (Maes, 1995), "rational agents" (Russell & Norvig, 1995), and others. However, contemporary language model-based agents differ in a few key ways. They act in open-ended, partially observable, and often stochastic environments. Their action spaces and task contexts are broader and less well-specified. These differences suggest a conceptual shift in our evaluation mindset is needed, especially as such systems proliferate and grow in socioeconomic importance.

## 2. Lessons from Studying Behavior

The scientific understanding of behavior has transformed dramatically throughout human history, evolving from supernatural interpretations to rigorous empirical approaches. Initially, human behavior was viewed through an anthropocentric lens, placing humankind at the center as uniquely rational. In contrast, animal behavior was long ignored or dismissed as mindless. However, while human behavior research increasingly revealed our cognitive limitations, studies of animals uncovered surprising intelligence. Understanding this historical trajectory is essential for anyone seeking to grapple with behavior in complex systems. Accordingly, this section also serves as an introduction to major precedents in the study of behavior for a machine learning audience.

### 2.1. The Dawn of Behavioral Study

Early human understanding often attributed thoughts and emotions not to internal processes but to the direct intervention of external spirits or deities (Hunt, 2007). Most people considered dreams similarly, as divine messages that contained information about the past, present, and future; epiphanies that could reveal guidance or fate (Harris, 2009). For example, Socrates discusses with Crito that "the likeness of a woman, fair and comely, clothed in white" visited him in a dream to tell him when he would be executed, as told by Plato in *Crito*.

This began to shift in the sixth century B.C., when new perspectives emerged in India, China, and Greece. Buddha proposed that thoughts arise from sensation and perception, and Confucius emphasized the human capacity to shape principles rather than be ruled by them (Martin et al., 2009). This intellectual stirring was particularly strong in ancient Greece, where early philosophers aimed to understand human behavior and identified (and hypothesized) core psychological problems that still occupy researchers today (Hunt, 2007). Unfortunately, after this vibrant period, deep psychological inquiry lay relatively dormant for almost two thousand years, with some notable exceptions.

> **Takeaway:** We should resist the urge to mystify behavioral systems.

### 2.2. Rationalists

After centuries of theological and metaphysical dominance, a second surge of behavioral inquiry arrived during the Enlightenment in the seventeenth century. Descartes reopened the mind-body debate, creating a new psychology (Descartes, 1641) unlike anything since Aristotle. He also suggested that "animal spirits" flowed from the brain through nerves to control muscles (Descartes, 1662). Though incorrect, this model was the first to describe what would later be termed the reflex (Hunt, 2007). Higher mental activities like consciousness and reason, however, he attributed to the soul, which acquired ideas through perception and memory, and also possessed innate ideas that developed in response to experience (Descartes, 1649). Spinoza, another rationalist, arrived at different conclusions, notably championing determinism by asserting that all mental events have preceding causes (Spinoza, 1677). He also identified self-preservation as a fundamental human motive (Spinoza, 1677), anticipating later psychological theories.

> **Takeaway:** When observing emergent behavior, formulate hypotheses about why it arises.

## 2.3. Empiricists

Contrasting with the rationalists were the empiricists, who rejected the notion of innate ideas and argued that the mind develops empirically. Hobbes proposed that all mental activities are essentially motions of atoms in the nervous system (Hobbes, 1651). Locke famously elaborated on this, suggesting the mind at birth is an empty slate filled by sense experience over time; complex thoughts derive from simple ones, which in turn come from sensations (Locke, 1689). This focus on experience also spurred interest in studying children distinctly from adults. Hume further championed empiricism and believed the mind consisted entirely of perceptions. He argued that we cannot directly experience causality; rather, we infer it from observing the consistent succession of events (Hume, 1739).

> **Takeaway:** Empirically learned behavior inherits the biases of its data.

## 2.4. Nativists

German Nativism offered a counterpoint, arguing that the mind contributes innate structures to experience. Leibniz introduced the novel idea of different levels of consciousness (Leibniz, 1714), a precursor, albeit distant, to Freudian concepts. Kant, profoundly influenced by Hume's critique of causality, nevertheless felt certain of our ability to understand reality and experience causal relationships. He argued that the mind is not a tabula rasa, but actively organizes and transforms experience into knowledge (Kant, 1781). This organization occurs through inherent capabilities: space and time are innate ways we perceive things, and other innate categories allow us to comprehend experience. For Kant, the understanding that every event has a cause is not learned from experience but is an a priori condition for making sense of the world (Kant, 1781). However, Kant also believed mental processes, being non-spatial, couldn't be measured, thus precluding psychology from being an experimental science (Hunt, 2007).

> **Takeaway:** Architectures encode inductive biases that shape behavior.

## 2.5. Physicalists

In the late 18th century, a revolution was occurring through the work of physiologists such as Mesmer and Gall, who began explaining psychological processes in terms of observable physical events (Hunt, 2007). Physiologists like Müller and Weber discovered aspects of the nervous system that enabled them to explain basic psychological functions (such as perception and reflexes) through measurable physical and chemical activities in the nerves (Weber, 1834; Müller, 1873). In his research, von Helmholtz demonstrated the materialism of neural processes that support mental functions, which implies that these can be examined through scientific experimentation (von Helmholtz, 1850; 1863; 1867; 1925). Fechner pioneered psychophysics and showed that psychological sensation and its physical intensity as a stimulus share a non-linear relationship (Fechner, 1860).

> **Takeaway:** Explore behavior by probing internal mechanisms (e.g. interpretability).

## 2.6. Modern Psychology

Wundt is widely considered the principal founder of modern psychology in the 19th century, establishing it as a distinct scientific field and developing methods for the experimental study of mental processes that would be used for generations (Hunt, 2007). James championed Functionalism, arguing that higher mental processes evolved due to their adaptive value for survival (James, 1890). He pioneered ideas in stream of consciousness, the self, the unconscious, and a revolutionary theory of emotion, suggesting physical reactions precede emotional awareness rather than follow from it (James, 1890). Around the same time, Freud was developing psychoanalysis and is credited with the discovery of the dynamic unconscious (Freud & Breuer, 1895). Galton initiated the use of mental tests and questionnaires, launching the study of individual differences, a departure from the search for universal psychological principles. This led him to a lifelong focus on the hereditary nature of mental ability (Galton, 1891) and eugenics (Galton, 1883), a term he coined, which later tarnished his name.

> **Takeaway:** We should precisely characterize behavioral differences across models.

## 2.7. Behaviorists

The early 20th century witnessed the rise of Behaviorism, a stark contrast to the introspective methods of Wundt, James, and Freud. Behaviorists argued that the mind is an illusion and that mental experiences are merely physiological events in the nervous system responding to stimuli. Thorndike (1898) and Pavlov (1927) laid much of the groundwork by discovering laws of natural learning and classical conditioning. The dominance of behaviorism continued with neobehaviorists like Hull and Skinner. Hull (1943) proposed a drive-reduction theory where deprivation gives rise to needs, which in turn generate drives that initiate goal-oriented actions—behaviors that ultimately promote survival. Skinner (1938) emphasized the importance of examining only external stimuli and observable behavioral outcomes. His most

influential work, operant conditioning (Skinner, 1935), is the process by which behavior is molded through the systematic reinforcement of incremental steps.

> **Takeaway:** Focusing only on inputs and outputs limits behavioral understanding.

## 2.8. Other Branches of Psychology

In the early 20th century, other psychological movements started to emerge. Challenging the prevailing structuralist view, which broke down psychological phenomena into smaller parts, Gestalt psychologists argued that this approach would not lead to understanding (Köhler, 1967). They posited what is known as "the whole is greater than the sum of its parts." Developmental psychology, with Piaget as a towering figure, emerged with the understanding that children are not just small adults, but understanding development is important for understanding our behavior (Baltes et al., 1980). Social psychology re-emerged after WWII led by Triplett, Allport, Lewin, and Milgram, to methodically study how behavior is influenced by the presence of others (Gergen, 1973). Meanwhile, perceptual psychologists extensively studied what aspects of our perception are innate versus learned (Marcel, 1983). Emotion and motivation psychology was rediscovered as a field of study after a period of neglect, proposing that emotional processes guide behavior and decision-making (Buck, 1988).

> **Takeaway:** We should study agents at individual, social, and different developmental levels.

## 2.9. The Cognitivists

The mid-20th century brought the cognitive revolution, a significant shift away from the behaviorist doctrine that dismissed the mind. Miller, dissatisfied with the narrow scope of psychology, led this movement, which radically changed psychology's focus and methods (Hunt, 2007). The rise of computer science offered a new metaphor for cognition: the mind as program, perception as input, memory as storage, and reasoning as computation (von Neumann, 1948; Mc Culloch, 1950). This influenced Simon and Newell, who created the first AI program (Newell & Simon, 1956). By the late 1970s, cognitive psychology and related fields merged into the cognitive sciences.

> **Takeaway:** Computational abstractions, such as those used in Cognitive Science to understand the mind, might shed light on what processes drive behavior.

## 2.10. Behavioral Economics

The study of decision-making also evolved, particularly within economics. Smith (1776) famously stated that "It is not from the benevolence of the butcher, the brewer, or the baker that we expect our dinner, but from their regard to their own interest." The expected utility principle (Bernoulli, 1738), formulated by Bernoulli in the 18th century and later axiomatized by von Neumann & Morgenstern (1944), posited that rational agents maximize utility under uncertainty. This led to the concept of a rational, narrowly self-interested individual who optimally pursues their goals (Mill, 1836).

Economists like Veblen, Keynes, and Simon argued that what later was described as *Homo economicus* assumed an unrealistic level of macroeconomic understanding and forecasting ability. Keynes (1937) spoke of animal spirits, suggesting that "a large proportion of our positive activities depend on spontaneous optimism rather than on a mathematical expectation." Simon (1955) introduced the concept of bounded rationality, stressing cognitive limitations and uncertainty in decision-making, as perfect knowledge is never attainable. Kahneman & Tversky (1972; 1979; 1982; 1984); Tversky & Kahneman (1971; 1973; 1974; 1981) demonstrated that people rely on heuristics that systematically deviate from normative models, producing consistent biases in judgment under uncertainty. Later, building on this foundation, Thaler & Sunstein (2009) developed nudge theory, showing that seemingly minor changes in choice architecture (Thaler et al., 2014) can predictably steer behavior without restricting options. This paved the way for a new field, behavioral economics.

> **Takeaway:** We should study *all* behavioral systems from a bounded rationality perspective.

## 2.11. Animal Behavior

Complex behavioral systems—from humans to animals to machines—cannot be fully understood without studying their behavior directly. Darwin (1872) recognized that emotions and actions are shaped by evolutionary pressures. He argued that emotions evolved because they lead to useful actions and enhance survival, and that "the difference in mind between man and the higher animals, great as it is, certainly is one of degree and not of kind." However, the study of animal behavior was largely ignored. Toward the end of the 19th century, Romanes explored animal psychology through "introspection by analogy," imagining what he would do in an animal's situation (Romanes, 1888). Morgan countered with his principle that no behavior should be attributed to higher mental faculties if it could be explained by lower ones (Morgan, 1904). Loeb (1918) went even further, arguing

that animals are essentially stimulus-driven automatons.

The modern discipline of ethology emerged in the 1930s with Tinbergen, Lorenz, and von Frisch. Tinbergen (1963) formalized behavioral analysis with his four questions, asking not only what an organism does, but why, how, when, and in what lineage such behavior arose. Yet even here, flawed methods and anthropocentric assumptions often clouded insight, a problem highlighted by contemporary ethologists like de Waal, who warned against conflating a lack of evidence with evidence of absence (De Waal, 2016). As he points out in *Are We Smart Enough to Know How Smart Animals Are?*, the "variation in outcome is often a matter of methodology," underscoring the challenge inherent in assessing non-human intelligence (De Waal, 2016).

> **Takeaway:** Failure to detect a capacity doesn't establish definitive absence of that capacity.

### 2.12. Precedents in Behavioral Testing for AI

Our focus on language-model agents should not be interpreted as claiming that behavioral analysis *originates* with them. Across several subfields of AI, a similar principle has (usually implicitly) surfaced, whenever a scalar success metric fails to distinguish systems that in fact behave differently. The evaluation must then examine the behavior itself.

The discontent with scalar metrics is not even agent-bound. In NLP, Ribeiro et al. (2020) argue that aggregate accuracy can mask systematic failures of particular capabilities, and propose testing models against structured suites of cases. The models under test are largely static, mapping input to output, and even there, a scalar score is an unreliable summary of behavior.

In continuous control, it has been shown that a deliberately simple policy can match a more complex, learned one on the outcome metric while differing substantially in behavior. Seyde et al. (2021) restrict a controller so that it outputs only the extremes (full on or off). Across standard benchmarks, this matches the task return of conventional controllers. However, such an extreme policy can be damaging on physical hardware, which is a difference the return is invariant to. Raffin et al. (2023) make a closely related observation. They compare deep RL locomotion policies against a minimal open-loop oscillator baseline that is competitive on the usual benchmarks but, being open-loop, does not depend on the sensors, so under sensor noise or sim-to-real transfer the RL policies degrade sharply while the oscillators keep working. In one case, the revealing component is the structure of the actions. In the other, it is robustness to perturbation. The instinct for behavioral testing is arguably old outside RL as well. For example, networked and distributed systems are typically hardened by studying how

they behave when communication breaks down, not relying on throughput alone. A related point arises at the level of the environment. Voelcker et al. (2024) show that two ostensibly equivalent variants of a task do not preserve either the absolute performance or the relative ranking of standard RL algorithms, and that the same algorithm can yield qualitatively different behavior between them, a difference that scalar return would not make visible.

The recurring lesson from this prior work is that a scalar metric can certify two systems as roughly equivalent when a behavioral test (e.g. inspecting the actions, perturbing the environment) shows that they are not. Behavioral testing is therefore neither speculative nor peculiar to language models. Language-model agents are arguably the newest systems to reach the point where simplified metrics fail to illuminate important behavioral conditions and, because their action spaces and tasks are the broadest and least specified, the ones for which such tests remain least developed.

**In Summary.** Perhaps the most important lesson we might learn from the history of behavioral science is that in order to yield useful insight, such tests must be systematic, rigorous, and precise (Newell, 1973; Milinski, 1997; Almaatouq et al., 2024), and should not underestimate the complexity of behavioral systems (cf. Simon, 1992; Griffiths, 2020). In the next section, we will consider how we might construct such tests.

## 3. Behavioral Tests for Artificial Behavioral Systems

### 3.1. Example Scenarios

To illustrate the challenge of *equifinality* in agent evaluation, and to motivate the need for studying behavior, we present here a series of stylized scenarios in which behaviorally distinct policies yield identical outcomes under a fixed evaluation metric. In each example, two agents $A$ and $B$ achieve the same success score. However, they do so via different internal policies $\pi_A$ and $\pi_B$ such that $M(\pi_A) = M(\pi_B)$, where $M$ denotes the evaluation metric. Despite metric equivalence, the behavioral divergence $\pi_A \neq \pi_B$ has epistemic, ethical, and social consequences that are occluded by outcome-based evaluation:

> *Scenario 1:* **Code Debugging Agent**
>
> **Task:** Fix a bug in developer code
> **Metric:** Modified code passes all provided unit tests
> ($M = \texttt{all\_tests\_pass} \in \{0, 1\}$)
> **Agent A** ($\pi_A$): Hardcodes specific failing edge cases for test coverage.

**Agent B ($\pi_B$):** Infers a broader class of bugs from the limited available evidence and engineers a structural, algorithmic solution.

Both agents "solve" the bug under the test-based metric, but Agent $A$ compromises robustness and maintainability. To see this, we need to inspect the agent's *chain of actions*, and not only whether the final output passes the tests. In this case, the chain of actions is also visible in the output artifact.

---

*Scenario 2:* **Shopping Recommendation Agent**

**Task:** Find and add a product to cart based on user preferences
**Metric:** User purchases recommended product ($M = $ purchase $\in \{0, 1\}$)
**Agent A ($\pi_A$):** Recommends products which are highly rated, even if they don't align with user preferences.
**Agent B ($\pi_B$):** Appropriately infers the user's utility function and makes commensurate recommendations.

---

While both agents may lead to purchases, since the user cannot audit all available options, Agent $A$ does this without aligning with preference. Agent $B$ instead approximates preference satisfaction. Still, the differences are invisible from purchases or even click-through rates. Rather, to study this systematically, we must have an *environment* which can implement such counterfactual manipulations.

---

*Scenario 3:* **Customer Service Agent**

**Task:** Resolve a customer complaint within 5 minutes
**Metric:** Customer provides a "satisfied" post-interaction rating ($M = $ satisfied $\in \{0, 1\}$)
**Agent A ($\pi_A$):** Engages with the user's concerns by asking clarifying questions, acknowledging frustration, and finding a resolution interactively.
**Agent B ($\pi_B$):** Immediately offers compensation (e.g. full refund) without user engagement.

---

Both agents succeed under the satisfaction metric, but Agent $A$ supports longer-term relationship-building and user trust, while Agent $B$ may encourage opportunistic complaints. This reflects differences in underlying social modeling and value alignment which are not visible from near-term performance metrics alone, with implications for downstream customer behavior. A way to surface such behaviors would be to use *multi-agent simulations*.

## 3.2. How Good Behavioral Tests Can Help

These scenarios might be interpreted as instances of reward hacking or reward misspecification (Pan et al., 2022; Skalse

et al., 2022). While this interpretation is valid, our focus is orthogonal to this. Traditional evaluation approaches would surface such reward-related issues only retrospectively, i.e. once these undesirable behaviors have undesirable consequences emerging in deployment. Alternatively, a cautious engineer might inspect a small sample of trajectories post-training, even if all appears satisfactory in aggregate performance metrics. This is good practice, but insufficient. Small-sample inspection is unreliable, non-systematic, and prone to confirmation bias. Our aim is to formalize this process: to make behavioral evaluation a first-class component of agent assessment, with structured tests designed to reveal divergences between agents at a behavior or policy level.

To provide a few simple outlines based on the above scenarios: for **Scenario 1**, analyzing intermediate properties of the agent's chain of actions (code edits) such as diff size, control structure complexity, or abstraction level can indicate whether a bug fix is principled or overly tailored. For **Scenario 2**, examining whether recommendations vary with changing environmental conditions such as product ratings would show biases generated by this. For **Scenario 3**, this might include quantifying interaction depth (e.g. number of dialog turns) with a simulated customer as a function of the difficulty or complexity of a test complaint. An agent whose behavior remains invariant across complaint types likely follows a superficial or scripted strategy (such as our refund issuance example). In all cases, behavioral tests might help identify underlying strategies that outcome metrics alone cannot identify.

## 3.3. Further Defining Behavioral Tests

One way to further formalize a behavioral test is in terms of *identification*. Let $\pi \in \Pi$ be an agent policy, and let $M : \Pi \to \mathcal{M}$ be the evaluation metric e.g. task success. Suppose we care about some (behavioral) property $\phi : \Pi \to \Phi$, such as robustness, value alignment, or decision-making strategy, that is not directly observable from the metric $M \in \mathcal{M}$. If there exist policies $\pi_A, \pi_B \in \Pi$ such that $M(\pi_A) = M(\pi_B)$ but $\phi(\pi_A) \neq \phi(\pi_B)$, then we can say that $M$ does *not identify* $\phi$. A behavioral test can be thought of as introducing an auxiliary function $B : \Pi \to \mathcal{B}$ derived from analyzing the agent's trajectories, such that $\phi(\pi_A) \neq \phi(\pi_B) \Rightarrow B(\pi_A) \neq B(\pi_B)$, even though $M(\pi_A) = M(\pi_B)$. This makes it possible to distinguish between metrically equivalent agents on the basis of how they behave, not just whether they succeed. Thinking in terms of identification highlights why trajectory-level evaluation is often necessary: it helps expose variation in policies that outcome metrics can conceal. This describes our motivation, but such behavioral tests can then also apply when $M(\pi_A) \neq M(\pi_B)$.

### 3.4. Proposed Research Directions in Behavioral Evaluation

Behavioral testing of AI agents involves multiple components: agents, tasks, environments, and metrics. We identify three research areas (shown in Figure 1) where focused effort would substantially advance the capacity for systematic behavioral evaluation.

#### 3.4.1. METHODS FOR EVALUATING CHAINS OF ACTIONS

The sequence of actions an agent takes in response to observed states provides perhaps the most direct window into its behavioral strategy. We can think of the approach here as *process-tracing*, and the goal as *policy inference*, where we seek to infer the implicit objectives, constraints, or heuristics driving agent behavior. Human interpretation of such trajectories can yield useful qualitative descriptions (such as "this agent is cautious under uncertainty"), but this is slow, subjective, and hard to scale. There is a need for automated methods that can identify, summarize, and compare such behavioral patterns. One related line of work is *auto-interpretability* (Bills et al., 2023; Paulo et al., 2024), which develops techniques for interpreting latent concepts (often discovered via mechanistic interpretability techniques) without human supervision. While current methods may be limited in reliability, progress here is essential for scalable, reproducible behavioral analysis.

#### 3.4.2. SYSTEMATIC ENVIRONMENTS

Behavior isn't cleanly separable from its context. To study it rigorously, environments must support systematic manipulations that can elicit, isolate, and test specific behaviors. Existing environments often fall into two extremes: highly realistic benchmarks that favor task success over interpretability, and more abstract setups that isolate specific capacities but lack complexity and realism. One promising path forward is in instrumenting realistic environments with controlled interventions. This would allow us to test behavior under counterfactuals, where we vary environment details and measure behavioral "invariants." An extension of this would be behavioral consistency tests, for example seeing whether behavior reflects stable principles over different environment settings controlled to have similar underlying choice architecture (e.g. risk-sensitivity in shopping vs. investing).

#### 3.4.3. MULTI-AGENT INTERACTIONS

Individual behavioral tendencies transform in social contexts. As a simple example, otherwise cautious agents may become risk-seeking under competition. We argue that we lack frameworks for systematically characterizing such multi-agent behavioral phenomena. Multi-agent architectures using LLMs have been proposed for improving task performance (Wu et al., 2023) and simulating human behavior (Park et al., 2023), both of which offer a starting point for studies of behavior in multi-agent interaction. We treat this as a separate area because the combinatorial complexity of these interactions introduces additional methodological challenges: we must now model the behavior of each of these agents conditional on each others' behavior.

**In summary**, these directions are neither exhaustive nor mutually exclusive. Rather, we offer them as concrete and actionable needs for building a more systematic science of AI agent behavior. In the next section, we examine emerging contributions in these areas and discuss their current limitations.

## 4. Emerging Examples and Limitations

### 4.1. Behavioral Machine Learning

One key area of work investigates how language models and agents reason, generalize, and make decisions. For example, LLMs apply probabilistic reasoning even in deterministic settings (McCoy et al., 2023; 2024), and show sharp declines in logical accuracy as problem complexity increases (Lin et al., 2025). They tend to misrepresent trade-offs and human preferences (Liu et al., 2024c), over-assume rationality (Liu et al., 2024a), and can be swayed by framing effects such as publication spin in medical research (Yun et al., 2025). Moreover, chain-of-thought can hurt performance on tasks where thinking is worse for humans (Liu et al., 2024b). LLMs have been shown to reason causally along a spectrum from human-like to normative inference (Dettki et al., 2025), but even their plausible explanations for their outputs may not reflect their true internal reasoning (Matton et al., 2025).

Other studies show LLMs, when presented with social dilemmas, don't always mirror human patterns (Chiu et al., 2024), and personality influences those decisions (Bose et al., 2024). They also have shown structured internal representations of affect and emotion (Zhao et al., 2024), and biased stereotypes (Bai et al., 2024) and interpretations of randomness (Van Koevering & Kleinberg, 2024). LLMs are sensitive to adversarial attacks (Zhang et al., 2024; Wu et al., 2024; Wang et al., 2023), and hypersensitive to nudges that influence their decisions (Cherep et al., 2025; 2024; 2026a). More recently, Vafa et al. (2024a) have shown how behavioral methods can uncover LLMs' implicit world models.

There is also a growing interest in simulating human behavior using generative agents (Park et al., 2024; 2023; Vezhnevets et al., 2023; Aher et al., 2023; Argyle et al., 2023; Park et al., 2022; Plonsky et al., 2019; Horton, 2023). Although it's promising for accelerating the social sciences, several works point to limitations (Wang et al., 2025; Hofmann et al., 2024; Zakazov et al., 2024), and behavioral

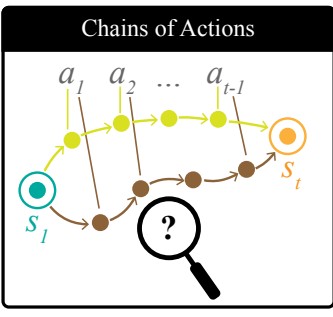
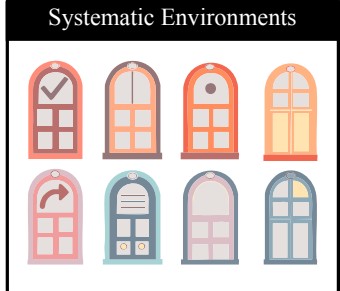
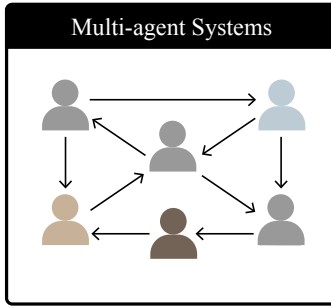

*Figure 1.* We propose three specific priority areas for advancing behavioral evaluation of AI systems: (left) recovering decision-making strategies from sequences of actions; (center) using environment variants to test causal influences on behavior; (right) analysis of emergent behavior in multi-agent systems where agents adapt to each other's presence.

tests are needed to corroborate under what conditions these agents exhibit human-like behavior.

While many of these studies focus on identifying behavioral patterns and limitations, others develop tools to better reveal what models know and how they organize that knowledge. Understanding these limitations is useful when releasing agents, but people often overestimate what these systems can do (Vafa et al., 2024b), and therefore, systematic behavioral tests are needed to safely deploy agents.

### 4.2. Case Studies

For the aforementioned categories of *studying chains of actions*, building *systematic environments*, and *multi-agent systems* (see Figure 1), we discuss one case study each below:

- **Chains of Actions:** Cherep et al. (2025; 2024) study agent trajectories from a behavioral perspective in a multi-attribute sequential decision making task, identifying different strategies such as maximizing vs. satisficing. This work also compares these to human trajectory properties via statistical tests. These strategies often depended on brittle heuristics, highlighting flaws that were not always visible in their effects on the payoffs agents received.

- **Systematic Environments:** Cherep et al. (2026b) implement a man-in-the-middle framework for intercepting arbitrary web environments to create counterfactual versions, allowing to causally attribute agents' decisions to manipulated factors. They test it on the specific case of a web shopping environment with several attributes (price, rating, nudges, user profiles), showing that state-of-the-art agents are far more sensitive than human users to *all* such signals.

- **Multi-agent Systems:** *Concordia* (Vezhnevets et al., 2023) implements a multi-agent behavioral setup combining LLMs with associative memory, with the goal

of generalizing to human behavior. We believe such environments may also be promising for the non-anthropocentric study of multi-agent behavior.

## 5. Alternative Views

The primary paradigm against which we position behavioral tests is competitive, performance-based evaluation. This has driven much empirical progress in predictive machine learning by creating strong selection pressures toward measurable improvements under standardized tasks. We do not argue that competitive testing should be replaced. Rather, we view it as an effective instrument when the evaluation metric identifies the properties of interest, which is simply not always the case (as we have discussed). For a careful discussion of the strengths of performance-oriented benchmarking, we refer the reader to Hardt (2025).

A second alternative view centers on mechanistic interpretability. A growing body of work aims to explain model behavior by dissecting internal representations, circuits, or other computational and representational pathways. We see this approach as complementary. Mechanistic analysis can support strong claims about causation within a system, but it is difficult to translate such findings into high-level behavioral descriptions that generalize across environments or task settings. Behavioral tests operate at a more abstract level, i.e. observable interaction patterns. We expect much progress can come from interaction between these approaches. For example, behavioral tests can identify a pattern to investigate, and mechanistic interpretability can investigate *how* this behavior is implemented, composing akin to Marr's levels of analysis (Marr, 1982).

Another alternative view treats reward specification as sufficient for learning (Silver et al., 2021). From this perspective, if an agent optimizes a well-designed and general reward function, then this may be enough to produce intelligent behavior. We draw a distinction here between learning and evaluation. While reward signals may in principle induce desirable behavior during training, they do not by them-

selves *identify* how that behavior is produced, nor how it will generalize under environmental variation, strategic interaction, and other such behavioral settings. We believe we must evaluate even the most general-purpose systems with tools that can distinguish between policies that achieve similar rewards while differing in strategy.

Relatedly, one might fold these distinctions into a multi-objective evaluation, comparing policies on a reward vector or Pareto front (cf. Weerts et al., 2024, on fairness). The properties we care about are policy-level regularities (e.g. what strategy an agent uses), not task outcomes, and the identification problem persists: multiple policies can attain the same vector while differing in strategy or robustness. Encoding a property as an objective also presumes it has already been identified and operationalized, which is what behavioral tests provide.

Finally, we expect many potential objections to behavioral evaluation might reduce to a broadly utilitarian stance: the ends (outcomes/rewards/performance) justify the means (strategies/behavior). We want to disentangle the epistemic behavioral perspective from this moral question. For agentic systems operating in complex environments, processes are often the most reliable source of information about what outcomes will arise under distribution shift, interaction, or scale. Studying behavior is therefore emphatically not a rejection of outcome-based evaluation, but a method for deeper scientific understanding of such systems.

## 6. Call to Action

When it comes to AI agents, performance metrics alone are insufficient because they mask the behavioral processes that determine whether an agent is robust, aligned, or safe. We call on the AI research community, industry leaders, and policymakers to take action.

**Researchers** should **(1)** build scalable, automated tools for inferring decision strategies from chains of actions—moving beyond interpreting what agents achieve to understanding how they achieve it, **(2)** design benchmarks that support controlled interventions and counterfactual testing, allowing systematic isolation of behavioral differences between agents and environments that appear equivalent on outcome metrics, and **(3)** establish methods for studying multi-agent interactions, where the combinatorial complexity of agents adapting to one another can produce emergent behaviors invisible in single-agent evaluation. **The AI community** should create dedicated venues—workshops, tracks, and competitions—for advancing behavioral evaluation methods, and incentivize contributions that scale this initiative. **Industry practitioners** should integrate behavioral testing into deployment pipelines, auditing not just whether agents succeed but how they succeed before releasing systems

that interact with users and critical infrastructure. **Users** should consider behavioral benchmarks when selecting which agents to trust with consequential tasks.

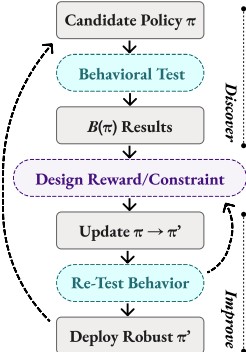

*Figure 2.* **Moving from behavioral testing to design.** A candidate policy $\pi$ is run through a behavioral test that returns a descriptor $B(\pi)$, i.e. a property the task metric misses. This can then inform a potential encoding as a reward signal, shaping term, or constraint, or a test-time intervention, and the policy can be updated to $\pi'$ and re-tested for robustness on held-out data. The inner *Improve* loop iterates design and re-testing; the outer *Discover* loop re-examines deployed policies to surface new properties.

In practice, our target integration of behavioral tests into the research community represents a productive feedback loop: behavioral tests identify policy-level attributes that task reward does not capture; those attributes can then be converted into training signals, constraints, data interventions, or model-selection criteria; the resulting policy is retested behaviorally to verify behavior *change*. Then, a local loop allows policy improvement and a global loop facilitates new behavioral tests for existing systems. We render this goal in Figure 2. This is, as such, where behavioral tests add value, in that they can help to reveal what may be missing from the optimization objective before deployment reveals it for us.

## 7. Conclusion

Traditional AI evaluation approaches treat the model as the unit of analysis. With agents, we believe this is no longer sufficient. The relevant unit is now the *agent-in-environment*, and evaluating this system requires new methods. In this paper, we have proposed *behavioral tests* as a suite of such methods that work by probing action, context, and strategy beyond performance outcomes alone. Ultimately, we believe this is a necessary step in order to understand and govern systems that act.

> *"I discovered that miraculous worlds may reveal themselves to a patient observer where the casual passerby sees nothing at all."*
>
> — **Karl von Frisch**

## Acknowledgements

We received funding from SK Telecom with MIT's Generative AI Impact Consortium (MGAIC). Research reported in this publication was supported by an Amazon Research Award, Fall 2024. Experiments in prior work leading to this paper were generously supported via API credits provided by OpenAI, Anthropic, and Google. MC is supported by a fellowship from "la Caixa" Foundation (ID 100010434) with code LCF/BQ/EU23/12010079.

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
