# OpenReview forum: "Position: Behavioral Systems Require Behavioral Tests"
_ICML.cc/2026/Position_Paper_Track — ICML 2026 Position Paper Track regular_

### Official Review · Reviewer_KWD8 · 2026-02-27

**Significance:** 4
**Argument Clarity:** 4
**Rating:** 5
**Confidence:** 5

**Questions:**

Q1. The paper treats behavioral testing as a post-hoc evaluation mechanism. How do you envision these behavioral tests ($B(\pi)$) being integrated into the optimization loop of agents in terms of shaping rewards and agent policies?

**Alternative Views Section:**

Yes

**Compliance With Llm Reviewing Policy A Conservative:**

Affirmed.

**Discussion Potential:**

4

**Final Justification:**

The rebuttal addressed my concerns.

**Paper Summary:**

The paper advocates for developing behavioral tests for agentic AI systems, ie evaluation strategies and environments that are not solely based on outcomes and task performance but consider the behavior of of such systems (policies and action trajectories and higher-level strategies and their invariances to perturbations). After providing a historical overview of studies of human and animal behavior, the authors argue given that AI agents are increasingly showing similar behavioral patterns, the ML community should also perform behavioral tests that analyze the interactions of AI agents with their environments and not the outcomes of such interactions. The authors then formally a behavioral test which can identify the agent's behavior and discriminate between two different behavioral trajectories that lead to the same outcome. Three axes are proposed for developing such behavioral tests; 1. analyzing chains of actions  and policies, 2. systematic environments to study behavioral invariance under perturbations, and 3. multi-agent environments and interactions. In the Alternative Views section, the authors position their approach as not contrasting but complementary to outcome-based evaluation (eg based on reward) and mechanistic interoperability (eg representational analysis).

**Position:**

Yes

**Position In Title:**

Yes

**Related Work:**

3

**Strengths And Weaknesses:**

## Strengths:

S1. The paper's position is very timely given that currently agentic evaluations and benchmarks are largely based on the final output and limited to task performance.

S2. By providing the background on studying behavior in psychology and ethology with takeaways for AI evaluation and the subsequent scenarios in which different behavioral policies for AI agents can lead to the same observable outcomes, the authors make their case well and justify why the ML community should consider developing behavioral tests and benchmarks for agentic evaluation.

S3. The formal definition of a behavioral test and the three proposed axes for behavioral evaluation provides a roadmap for ML researchers to design behavioral tests for AI agents.

S4. The authors consider two alternative views (pure outcome-based evaluation and mechanistic interoperability based on representations) and admit that their position is not in favor of abandoning these techniques but complementing them with behavioral tests to better understand AI agents' behavior.

## Weaknesses:

W1. While the authors advocate for behavioral tests as an evaluation paradigm, they fail to address how these insights can be systematically integrated back into the design or training of models. In that regard, there is a disconnect from the optimization loop of AI agents.

**Support:**

4

---

> ### Author Rebuttal · Authors · 2026-03-30
>
> Thank you for noting the timeliness of the work, the compelling justification for behavioral testing, the actionable evaluation axes, and the constructive framing of behavioral tests as complementing rather than replacing existing methods.
>
> > **While the authors advocate for behavioral tests as an evaluation paradigm, they fail to address how these insights can be systematically integrated back into the design or training of models.**
>
> > **Q1: The paper treats behavioral testing as a post-hoc evaluation mechanism. How do you envision these behavioral tests () being integrated into the optimization loop of agents in terms of shaping rewards and agent policies?**
>
> We agree that the current draft over-emphasizes the post-hoc evaluation component. In practice, the target integration of behavioral tests into the research community represents a productive feedback loop:
>
> 1. Behavioral tests identify policy-level attributes that task reward does not capture
> 2. Those attributes can then be converted into training signals, constraints, data interventions, or model-selection criteria
> 3. The resulting policy is retested behaviorally
>
> This is, as such, where behavioral tests add value in that they can help to reveal what may be missing from the optimization objective before deployment reveals it for us. We will describe this more directly.

---

> > ### Author Rebuttal · Reviewer_KWD8 · 2026-04-01
> >
> > I thank the authors for their response. I retain my score.

---

### Official Review · Reviewer_aG54 · 2026-03-08

**Significance:** 2
**Argument Clarity:** 2
**Rating:** 5
**Confidence:** 4

**Questions:**

* Why did the work not discuss examples from other agent based research?
* When does a multi-objective approach offer a better alternative to use of behavioral tests?

**Alternative Views Section:**

Yes

**Compliance With Llm Reviewing Policy A Conservative:**

Affirmed.

**Discussion Potential:**

2

**Final Justification:**

The rebuttal addressed my concerns and I have increased my score accordingly to an accept

**Paper Summary:**

The work argues that AI researchers should broaden their evaluation practices to not only focus on performance metrics (such as success metrics, accuracies, ...) but to additionally include behavioral tests. This is mainly motivated by the fact that agentic systems are becoming more prevalent and these systems do not produce a single outcome but (re)act in an environment until a final outcome is achieved. While to strategies might achieve the same final outcome (and performance metric value) the paths they take to get there might be vastly different. The authors provide toy examples to highlight such situations and argue that behavioral tests provide the means to uncover such differences. The work further provides an overview of how behavior has been studied and provides key takeaways from different eras.
The work centers this position around LLM-based agents and does not discuss other agent systems at large.

**Position:**

Yes

**Position In Title:**

Yes

**Related Work:**

3

**Strengths And Weaknesses:**

To me, the biggest weaknesses of this work are the strong focus on LLM-based agents and limitations to artificial examples. I do understand the authors argument that such systems are too narrowly evaluated with single performance metrics in mind. However, I believe the work would be much stronger if it would discuss how other agent systems, e.g., from robotics, RL, AI planning, networking or operating systems have already successfully made use of behavioral tests. For example, in multi-agent systems (be it AI planning, networks, or operating systems) communication protocols are often used to ensure all agents know essential information. Such communication protocols often are developed based on observed behavior of failed communication. While behavior testing might not have been a first-class citizen in the development of such protocols, I would argue that this is a clear case where evaluating the behavior resulted in more robust and reliable systems.
In RL researchers regularly compare learned policies to potentially simpler policies to gain better insights on how these policies differ. Somewhat recently, based on the insight that continuous control agents often favor max-and-min action values [arxiv.org/abs/2111.02552](https://arxiv.org/abs/2111.02552) evaluated if directly forcing such bang-bang behavior is beneficial in well known continuous control tasks. Counter to this https://rlj.cs.umass.edu/2024/papers/RLJ_RLC_2024_18.pdf shows that for many of the locomotion tasks RL aims to solve, simple oscilators, with drastically fewer (learned/tuned) parameters serve as important baselines and should not be overlooked. https://arxiv.org/abs/2410.08870 discusses that small differences in simulation can have drastic effects on the outcome for the same task.
I believe that this highlights that limiting the paper to artificial examples and focus on LLM-agents drastically reduces the utility of the paper. As the goal of the paper is to advocate for behavioral tests, particularly in the LLM-agent setting it would be good to highlight how different agent systems have already utilized behavioral analysis to improve their respective fields and how one could draw parallels to the novel agentic systems.

When defining behavioral tests (Section 3.3) I am not sure why the same goal can not simply be achieved with multi-objective evaluation.
For example, take two objectives to be performance and fairness. Both of them might use metrics that only look at the final outcome. Since they measure different quantities, they can help in differentiating policies from each other, even if they have the same performance/fairness. With multi-objective optimization, it is even possible to create a pareto front and look at the tradeoff between multiple objectives (see e.g. https://arxiv.org/pdf/2303.08485). I believe the work would be strengthened if it discusses of why behavioral tests are needed and why simply using multiple objectives often does not suffice.

The work otherwise presents a clear position that can have a clear impact on LLM-based agentic systems.


I am willing to increase my score if a more broader discussion on agent systems in general is included.

---
Minor Comments:
* Line 71: Shouldn't "undeserved" be changed to "unde ***r*** served"?
* Some bracketed citations should rather be text citations (see, e.g., right column Lines 152&153)

**Support:**

2

---

> ### Author Rebuttal · Authors · 2026-03-30
>
> Thank you for recognizing the clarity of the position and its potential impact on LLM-based agentic systems.
>
> > **To me, the biggest weaknesses of this work are the strong focus on LLM-based agents and limitations to artificial examples. I do understand the authors argument that such systems are too narrowly evaluated with single performance metrics in mind. However, I believe the work would be much stronger if it would discuss how other agent systems [...]**
>
> See Q1 below
>
> > **When defining behavioral tests (Section 3.3) I am not sure why the same goal can not simply be achieved with multi-objective evaluation. [...]**
>
> See Q2 below
>
> > **Minor Comments**
>
> Thank you for catching these!
>
> > **Q1: Why did the work not discuss examples from other agent based research?**
>
> Our intention was not to suggest that behavioral analysis is unique to LLM-based agents. We believe LLM-based agents are currently the setting in which the gap between behavioral richness and evaluation practice is most acute, and therefore that LLM-agent evaluation now requires the same behavioral maturity that other agent literatures have operationalized to different degrees.
>
> To this end, we believe the examples you mention support the paper's core claim i.e. once systems are sequential, interactive, and sensitive to context, scalar success metrics stop being sufficient for rigorously understanding what policy has actually been learned.
>
> We agree that making these precedents from the study of embodied agents explicit would strengthen the paper, and we will revise accordingly with these helpful examples you suggest; thank you for including these. We will use most of our extra page allowed in the camera-ready to tackle this.
>
> > **Q2: When does a multi-objective approach offer a better alternative to use of behavioral tests?**
>
> A multi-objective approach is preferable when:
>
> 1. The relevant distinctions can be specified ex ante as objective components, and
> 2. Policies that achieve the same objective value or vector can be treated as equivalent for the purpose at hand
>
> Our argument is that this condition often fails for agentic systems. Many properties we care about are not naturally task-level outcomes, but policy-level regularities, such as what cues the agent relies on, how brittle it is to perturbation, what strategy it uses, how it trades off near- and long-horizon considerations, or how it communicates in interaction with others. These properties can persist across many tasks and environments, and can potentially also help or hurt performance or robustness depending on context. Treating them purely as reward terms therefore conflates two separate questions: what objective is used in a given task, and what behavioral pattern the policy actually learns.
>
> Additionally, encoding a behavioral property into a multi-objective reward already assumes that the property has been correctly identified and operationalized, but this is precisely the role behavioral tests play. They are needed to discover, validate, and stress-test policy-level descriptors before those descriptors can be safely turned into objectives and other implements (constraints, regularizers, or model-selection criteria). In that sense, multi-objective training and evaluation can absorb insights from behavioral testing, but it does not eliminate the need for behavioral testing in the first place.

---

> > ### Author Rebuttal · Reviewer_aG54 · 2026-04-01
> >
> > See my comment above

---

### Official Review · Reviewer_TZiQ · 2026-03-12

**Significance:** 3
**Argument Clarity:** 3
**Rating:** 4
**Confidence:** 4

**Questions:**

The paper emphasizes the need to evaluate behavioral processes beyond outcome-based metrics. However, related work in reinforcement learning has explored multi-objective reward formulations and reward shaping to incorporate intermediate processes into evaluation. The paper could more clearly clarify how the proposed behavioral evaluation paradigm differs from or complements such approaches.

**Alternative Views Section:**

Yes

**Compliance With Llm Reviewing Policy A Conservative:**

Affirmed.

**Discussion Potential:**

2

**Final Justification:**

Since my questions have been resolved. I have raised my score by one level.

**Paper Summary:**

The paper proposes Behavioral Machine Learning as a research paradigm for studying how AI agents reason, generalize, and make decisions. It argues that evaluating agents solely through outcome-based performance metrics is insufficient, and instead calls for the systematic design and use of behavioral tests to analyze agent behavior. Through such tests, the paper emphasizes the need to evaluate agents’ strategies, adaptability, vulnerabilities, and interactions with other agents.

**Position:**

Yes

**Position In Title:**

Yes

**Related Work:**

3

**Strengths And Weaknesses:**

Strengths
- (1) Research on AI agents is an important and rapidly emerging topic for the ICML community. The paper highlights the need for a new research paradigm—Behavioral Machine Learning—to study agent behavior beyond outcome-based evaluation metrics.
- (2) The paper identifies three research categories (Chains of Actions, Systematic Environments, and Multi-agent Systems) and discusses current limitations and open challenges in each area. It also provides concrete case studies that help illustrate these issues.
- (3) Section 2 provides a broad survey of the historical development of behavioral sciences, offering useful context for positioning behavioral evaluation as a scientific framework for studying AI agents.

Weaknesses
- (1) Although the paper outlines a research agenda for behavioral evaluation, it does not provide sufficiently concrete methodologies, evaluation protocols, or experimental designs for implementing behavioral tests in practice.
- (2) Several related research directions—such as interpretability studies or work on reward design (e.g., multi-objective or multi-reward formulations)—have already discussed similar topics. This paper could more clearly articulate its novelty and distinguish its contribution relative to these existing lines of work.

**Support:**

3

---

> ### Author Rebuttal · Authors · 2026-03-30
>
> Thank you for appreciating the relevance of the proposed paradigm, the structured identification of research categories and open challenges, and the breadth of the historical context provided.
>
> > **Although the paper outlines a research agenda for behavioral evaluation, it does not provide sufficiently concrete methodologies, evaluation protocols, or experimental designs for implementing behavioral tests in practice.**
>
> We appreciate this request. One canonical workflow we have in mind for a typical analysis is (with examples from the nudge sensitivity work cited in the paper):
>
> 1. Identify a property of interest not captured by the task metric (e.g. nudge sensitivity)
> 2. Define a measurable trajectory- or interaction-level descriptor for that property (e.g. probability of choosing option A vs. B is A is nudged)
> 3. Construct controlled perturbations or task variants that should and should not change that descriptor (versions of A vs. B choice sets with either and neither nudged)
> 4. Compare agents/policies on the stability and sensitivity profile of that descriptor (P(choose A if nudged) \- P(choose A if not nudged) for each agent/policy)
>
> We will outline this more explicitly.
>
> > **Several related research directions—such as interpretability studies or work on reward design (e.g., multi-objective or multi-reward formulations)—have already discussed similar topics. This paper could more clearly articulate its novelty and distinguish its contribution relative to these existing lines of work.**
>
> We view these directions as complementary but not redundant:
>
> * Interpretability usually seeks internal explanations, e.g. which representations or circuits support a computation, but does not by itself identify policy-level regularities across environments or interactions
> * Reward design seeks better optimization targets, but does not by itself guarantee that the learned policy exhibits the intended strategy, since multiple strategies may achieve similar rewards under any fixed design
> * Benchmarking compares final task performance, but often cannot distinguish metrically similar agents that behave differently under perturbation or deployment (e.g. subsequently-ranked leaderboard models).
>
> Behavioral tests are designed for the residual structure here: they measure agent-environment regularities at the level of trajectories, sensitivities, invariances, and interactions.
>
> This provides for a potentially productive feedback loop, wherein interpretability can help understand how behaviors are implemented in the model, reward design can allow the control of behaviors based on identified patterns, and benchmarks can help isolate which the best performers at a task are whose behaviors should be compared.
>
> > **The paper emphasizes the need to evaluate behavioral processes beyond outcome-based metrics. However, related work in reinforcement learning has explored multi-objective reward formulations and reward shaping to incorporate intermediate processes into evaluation. The paper could more clearly clarify how the proposed behavioral evaluation paradigm differs from or complements such approaches.**
>
> We agree that multi-objective reward design is one important way to *optimize* for behavioral desiderata, but it does not remove the need to measure behavior separately. The reason is that behavioral properties are often properties of the policy itself, invariant to the final outcome: the same strategic tendency can help or hurt across different tasks and environments (for example, a model that has a general tendency to over-rely on superficial environment cues such as marketing slogans might do worse in many environments, but then better in one where those cues correlate with the quality of observed states). Treating such properties purely as reward terms can therefore entangle the scientific characterization of agent behavior with task-specific optimization.
>
> Finally, shaped and multi-objective rewards remain vulnerable to the same identification problem motivated in the paper: multiple policies can achieve similar (multi-)reward vectors while differing in strategy, robustness, or interaction style.

---

> > ### Author Rebuttal · Reviewer_TZiQ · 2026-04-03
> >
> > Thank you for your clarification. I will consider in final decision.

---

> > > ### Author Response · Authors · 2026-04-05
> > >
> > > Thank you! We hope you consider raising the score since you mentioned that all your concerns have been fully resolved.

---

### Official Review · Reviewer_dDm6 · 2026-03-12

**Significance:** 3
**Argument Clarity:** 4
**Rating:** 5
**Confidence:** 4

**Questions:**

1. How should behavioral evaluation be balanced with performance-based evaluation when the primary goal of a system is task performance?
2. How could the proposed behavioral evaluation framework inform the design of learning objectives or reward functions during training?
3. How might the proposed framework support generating or discovering new behavioral capabilities during training rather than only diagnosing existing behavior?
4. The paper emphasizes behavior that generalizes across environments. How should such behavioral descriptions be defined and measured in practice?

**Alternative Views Section:**

Yes

**Compliance With Llm Reviewing Policy A Conservative:**

Affirmed.

**Discussion Potential:**

4

**Final Justification:**

The paper presents a clear and well-motivated position advocating behavioral evaluation as an important component of AI system assessment, with intuitive examples and a timely focus on agentic systems based on large language models. The proposed framework is clearly articulated, well-positioned relative to related work, and the paper is well organized and written. The concerns regarding its connection to performance objectives, reward design, and implications for training are addressed in the rebuttal, which clarifies its complementary role and improves the overall positioning.

**Paper Summary:**

The paper argues that AI agents should be evaluated in a manner similar to other behavioral systems, through systematic observation, perturbation, and interpretation of their actions. The authors focus on agentic systems built on large language models and argue that current evaluation approaches are insufficient for understanding their behavior. The paper highlights that different internal policies may produce the same observable outcomes under standard evaluation metrics, which can obscure important behavioral differences. To address this, the authors advocate studying AI systems at multiple behavioral levels and propose systematic behavioral evaluations to better understand how agents interact with their environments. The paper positions this perspective as complementary to existing approaches such as interpretability, formal verification, and benchmarking.

**Position:**

Yes

**Position In Title:**

Yes

**Related Work:**

4

**Strengths And Weaknesses:**

**Strengths:**
- The paper clearly states a position advocating behavioral evaluation as an important component of AI system assessment.
- The motivation is well illustrated through intuitive examples showing how different policies can produce identical outcomes.
- The focus on agentic systems based on large language models is timely and relevant to current developments in AI.
- The proposed conceptual framework and formalism for behavioral analysis are clearly presented.
- The discussion situates the proposal relative to related approaches such as interpretability and reward hacking.
- The paper is well organized, clearly written, and likely to stimulate discussion within the research community.

**Weaknesses:**
- The paper could further clarify how behavioral evaluation interacts with performance-driven objectives in AI systems.
- The connection between behavioral evaluation and reward design or learning objectives is not fully explored.
- The implications for training or shaping agent behavior through the proposed evaluation framework remain somewhat unclear.

**Detailed Review:**

The paper presents a clear and well-motivated argument that evaluating AI agents purely through outcome-based metrics can overlook important behavioral differences. The examples illustrating how distinct strategies can lead to identical observable outcomes help clarify the concept of equifinality and demonstrate why behavioral evaluation may be necessary. The focus on LLM-based agentic systems is timely, and the framing of behavioral evaluation as complementary to interpretability, formal verification, and benchmarking is helpful.

The scenarios provided in the paper are effective in illustrating the evaluation challenges. In particular, the discussion of different behavioral strategies leading to the same result highlights the limitations of fixed evaluation metrics. The historical context and references to behavioral sciences provide useful background for readers and help situate the proposed perspective within a broader research tradition.

At the same time, several aspects could be further clarified. It remains somewhat unclear how behavioral evaluation would interact with systems whose primary goal is performance optimization. In addition, while the paper emphasizes identifying behavioral differences, it would be useful to discuss how these insights could inform training objectives or reward design. The relationship between reward shaping and behavioral evaluation could be explored more explicitly. Some sections, such as the discussion of reward hacking, may also benefit from additional references to existing literature.

Overall, the paper provides a thoughtful and well-organized perspective on evaluating AI behavior. The proposed formalism and diagnostic tests appear promising as tools for systematic behavioral analysis. The suggested research directions and call to action are clearly articulated and could guide future work in this area.

**Support:**

4

---

> ### Author Rebuttal · Authors · 2026-03-30
>
> Thank you for recognizing the clarity of the position, the intuitive motivation, the timeliness of the focus on agentic systems, and the contributions of the conceptual framework, related work discussion, and overall organization.
>
> > **Q1: How should behavioral evaluation be balanced with performance-based evaluation when the primary goal of a system is task performance?**
>
> We view behavioral evaluation as a complementary layer that becomes essential when performance metrics do not identify the properties we care about (e.g. robustness, alignment, strategy). In standard performance-oriented settings, this can be implemented as a lexicographic-style decision rule wherein we first require a threshold of task performance, and then use behavioral analysis to disambiguate the remaining policies.
>
> This is arguably the regime where behavior matters most, i.e. where multiple agents can solve the task, but differ in robustness, manipulability, social strategy, etc. in ways the task metric does not reveal.
>
> > **Q2: How could the proposed behavioral evaluation framework inform the design of learning objectives or reward functions during training?**
>
> > **The relationship between reward shaping and behavioral evaluation could be explored more explicitly. Some sections, such as the discussion of reward hacking, may also benefit from additional references to existing literature.**
>
> We agree that this can be stated more explicitly. A role of behavioral tests is to expose what the task reward is failing to identify. Once a test reveals a recurrent behavioral failure, that can be fed back into training in several standard ways: as an auxiliary objective, a constraint, a source of counterexamples for curriculum design, a model-selection criterion, etc. Put differently, reward design and behavioral evaluation can cooperate in a productive feedback loop that refines the performance and behavior of the underlying system.
>
> As a concrete example, some of the prior work we discuss identified nudge sensitivity via behavioral evaluation. A natural extension is to explore nudge-robust (post-)training objectives, such as stability of reward under nudged perturbations (for a simple example).
>
> > **Q3: How might the proposed framework support generating or discovering new behavioral capabilities during training rather than only diagnosing existing behavior?**
>
> Thank you for this excellent point: we do not see behavioral tests as only post-hoc diagnostics. Behavioral tests can indeed be constructive when they are used to search for missing invariances or missing strategies, or even to understand strategies e.g. in a validation loop during training (as an augmentation to manual review of agent trajectories).
>
> > **Q4: The paper emphasizes behavior that generalizes across environments. How should such behavioral descriptions be defined and measured in practice?**
>
> Behavioral objects can exist at multiple hierarchical levels. For example, a strategy can be as low level as "when uncertain about which line contains the needed information" $\to$ [strategy 1: "read the entire doc"] vs. [strategy 2: "use a grep tool to search the doc for keywords"]. It could also be as high-level as "if the other agent defects, then defect until they cooperate" (the classic tit-for-tat strategy from game theory). In practice, we anticipate that individual behavioral tests will need to state the kinds of objects they are seeking and formalize how this can be turned into a measurable descriptor over e.g. agent trajectories, and then test whether that descriptor is stable across different conditions.

---

> > ### Author Rebuttal · Reviewer_dDm6 · 2026-04-03
> >
> > Thank you for addressing my concerns. I will retain my current score.

---

### Decision · Program_Chairs · 2026-04-30

**Decision:**

Accept (regular)

**Comment:**

There is general consensus among the reviewers about the interest of the position expressed in the paper and how the authors articulated it. While I recommend acceptance for the paper, I would like to encourage the authors to properly integrate the feedback from the reviews and the rebuttal period, in particular about the following points:

* Connection with fields directly studying aspects related to behaviors and interaction. Rev. aG54 articulated very well this aspect and provided valuable references for the authors to complement their literature review and better position their position. Rev. TZiQ also mentioned connections with eg, interpretability and multi-objective rewards in RL, that should be further clarified. This feedback does not dismiss the value of the position expressed in the paper, but it would allow a more comprehensive review of different approaches that include behaviors/interaction as a primary component of evaluation.
* Clarify how behavioral evaluation protocols would ultimately contribute to designing/improving the algorithm (see Rev. dDm6 Q2-3, Rev. KWD8 W1). While this is a position paper and not full technical contribution, it is still important to clearly show what are the potential benefits of pursuing the ideas in the paper to motivate the community.